# Emerging Roles of RNA-Binding Proteins in Neurodevelopment

**DOI:** 10.3390/jdb10020023

**Published:** 2022-06-10

**Authors:** Amalia S. Parra, Christopher A. Johnston

**Affiliations:** Department of Biology, University of New Mexico, Albuquerque, NM 87131, USA; sancham9@unm.edu

**Keywords:** neuroblast, RNA-binding protein, neural stem cell

## Abstract

Diverse cell types in the central nervous system (CNS) are generated by a relatively small pool of neural stem cells during early development. Spatial and temporal regulation of stem cell behavior relies on precise coordination of gene expression. Well-studied mechanisms include hormone signaling, transcription factor activity, and chromatin remodeling processes. Much less is known about downstream RNA-dependent mechanisms including posttranscriptional regulation, nuclear export, alternative splicing, and transcript stability. These important functions are carried out by RNA-binding proteins (RBPs). Recent work has begun to explore how RBPs contribute to stem cell function and homeostasis, including their role in metabolism, transport, epigenetic regulation, and turnover of target transcripts. Additional layers of complexity are provided by the different target recognition mechanisms of each RBP as well as the posttranslational modifications of the RBPs themselves that alter function. Altogether, these functions allow RBPs to influence various aspects of RNA metabolism to regulate numerous cellular processes. Here we compile advances in RNA biology that have added to our still limited understanding of the role of RBPs in neurodevelopment.

## 1. Introduction

The complex question of how a nervous system is built is a fundamental topic in developmental biology. Neurogenesis is one of the earliest events in development that occurs during embryogenesis. During this process, existing cells give rise to new cells that acquire specific identities and move to different locations to begin building precursory tissues within the nervous system (i.e., brain, spinal cord, nerves). This is conducted in a remarkably organized manner to ensure that resulting tissues are of the correct size and have the structure necessary for function. Each cell type that makes up the diverse brain regions has unique transcript profiles, which influences cell identity, proliferation, apoptosis, and differentiation [1,2,3,4]. This helps create the correct number and size of cells with the desired properties and functionalities. The mechanisms by which these unique profiles are established are not clearly understood. In recent years, elegant studies have identified new components and mechanisms that regulate formation of the central nervous system (CNS) [5,6]. Still, several questions surrounding both normal and aberrant development of the CNS remain. One reason for these persistent unknowns is that there are various ways in which a cell can regulate its genetic profile, including transcriptional, hormonal, and metabolic regulation. This raises an important question: if cells in a living organism contain the same genetic material, then how does a cell know when and how to express or repress certain genes? Mechanisms of doing so differ across cell types and each cell must make efficient use of available energy to drive certain pathways. Among them, RNA-binding proteins (RBPs) provide an efficient and timely method to regulate gene expression profiles at the posttranscriptional level. This is performed through direct RNA binding, along with various interactions between RBPs and other gene expression regulators. RBP–RNA interactions can be transient, they can bind early and remain bound until degradation, or they can bind to influence downstream processing such as splicing or transport [7,8,9]. Together with the large number of RBPs estimated in the genome (~1500, [10]), these molecular properties yield an impressive degree of regulation on cell function. Not surprisingly, RBPs have emerged as critical regulators of development.

Here we describe recent advances in developmental neurobiology that contribute to our understanding of the multifaceted processes involved in brain development, focusing on the roles played by RBPs in neural stem cell function and maintenance. This family of proteins is involved in a plethora of cellular processes including localization, splicing, and translation of target RNAs. The wide array of RBP targets, including mRNA, miRNA, and lncRNA, allows amplified regulation of many cellular processes by RBPs, especially in stem cells [11,12]. Additionally, they allow for compartmentalized and temporal regulation of target metabolism. Learning about these processes will aid in understanding how the complex regions of the brain are formed from a relatively small pool of progenitor cells and how they function. To aid in this task, developmental biologists have turned to simple and easy-to-genetically-manipulate organisms such as *Drosophila* and *C. elegans*. Studies in these organisms can easily be reiterated in mammalian systems due to the highly conserved nature of many genes and pathways.

## 2. Neural Stem Cell Development

Early neurogenesis includes the formation of neural cells from ectodermal cells that, in turn, give rise to structures that become the brain, spinal cord, and peripheral nerves. The brain develops from a small pool of stem cells that are programmed to self-renew or differentiate to form an integrated and functional nervous system. Despite their relative low abundance, a high proliferative capacity allows these neural stem cells to continuously self-renew while also generating differentiated progeny throughout development. Should these properties go awry, proper development of an organism can be interrupted. For example, mutations in certain stem cell regulatory pathways have been shown to promote hyperplastic growth, whereas others lead to a decrease in the stem cell pool [13,14,15]. These events may lead to problems later in development and could contribute to several diseases including cancer. The stem cell populations in the brain have become of particular interest because of their potential to treat nervous system disorders as well. This, however, is hindered by the limited knowledge of how neural stem cells are regulated throughout development. In subsequent paragraphs we give a brief overview of neural stem cell development followed by specific studies highlighting the role of RBPs in CNS form and function. 

Neural stem cells arise during gastrulation of early embryonic development. These cells will go on to produce all the specialized cell types found in the brain and nervous system. In vertebrates, migratory stem cells, called neural crest cells (NCC), are involved in formation of cephalic structures including ganglia and skeletal structures [16,17]. This process requires accurate and intricate signaling between tissue environments, cells, and proteins [18]. The neural tube is the first structure generated by a series of complex molecular processes and gives rise to the brain and spinal cord [19]. Additional segments and specialized regions arise from the neural tube later in development, and these also rely on various molecular mechanisms. RBPs are important during development of these embryonic and fetal neural structures due to the rapid and simultaneous changes. Cells must express the correct genetic profile and molecular machinery to produce the necessary factors to differentiate or self-renew. These processes, amongst others highlighted in this review, are regulated by RBPs and discussed in detail in subsequent sections. 

*Drosophila melanogaster* (*D. melanogaster*), the common fruit fly, is a frequently used and established genetic model organism in developmental biology. The various stem cell populations in the fly can be studied across developmental time, and many developmental processes in humans can be easily recapitulated in flies. This has made the fruit fly an attractive model to study stem cells in disease and dysfunction. More specifically, neural stem cell (NSC) biology is often studied in *Drosophila* because the fruit fly brain contains a fixed population of neural stem cells (neuroblasts) that can be tracked throughout development. Moreover, development of some neuroblasts closely resembles that of neural stem cells, called radial glial cells (RGCs), in the ventricular zone (VZ) and subventricular zone (SVZ) of the mammalian brain where they generate neurons, oligodendrocytes, and astrocytes [20,21,22,23,24,25,26]. Like the neural stem cells in the mammalian brain, *Drosophila* neuroblasts undergo ACD to produce molecularly and physically distinct progeny [22,27,28]. By this mechanism, a small pool of neuroblasts can give rise to a large population of distinct, specialized cell types that comprise the complex adult brain while maintaining their own population through self-renewal. Neuroblast development relies on temporal and spatial expression of factors assigned at birth that dictate production of a specific subset of neurons/glia [29]. In the subsequent sections we will discuss these factors and how RBPs can regulate their activity.

## 3. Regulation of Spatial Gene Expression

Compartmentalization makes cellular processes and tissue functions more efficient. Throughout development, individual cells and tissues require spatial regulation of specific processes, notably gene expression, RNA processing, and translation, for example. Spatial regulation of gene expression is particularly important during tissue development. As cell identity is controlled by differential gene expression, stem cells rely on spatial regulation to correctly balance self-renewal and differentiation of their mitotic progeny. An important mode of regulation extends from localization of cognate mRNA to discrete areas within the cell that allow for differential inheritance during cell division. The precise mechanisms of RNA transport are still being elucidated, but several interesting details have been revealed [30,31,32], and we will only mention some commonalities here. Transport signals can be found at the 3′ UTR of mRNA [33], or they can be *cis*-acting motifs called zipcodes [34] that facilitate formation of a transport complex consisting of molecular motors and RBPs [35,36]. Binding of RBPs to the 3′ UTR is a common regulatory mechanism observed in development. For example, CUGBP ELAV-like family member-2 (CELF2) is an RBP that transports target mRNA between the cytoplasm and the nucleus. This ultimately serves as a mechanism to promote neural progenitor cell self-renewal or differentiation in the murine cerebral cortex [37]. Nuclear-cytoplasmic translocation can occur through formation of complexes that contain molecular motors. The zipcode-binding protein (ZBP1) interacts with a kinesin-I motor complex (Figure 1) in neurons to transport mRNA during murine postnatal development [38]. ZBP1 is closely related to insulin-like growth factor 2 mRNA-binding protein 1 (IGF2BP1) in mammals (Imp; *Drosophila*) (Table 1). IGF2BP1/Imp is involved in RNA localization and translation [39,40]. In *Drosophila* Imp, the prion-like domains (PLD) regulate formation and homeostasis of neuronal ribonucleoprotein (RNP) granules in axons [41]. Furthermore, Imp regulates *Drosophila* egg chamber development by regulating Notch activation. More specifically, Imp directs localization of the metalloprotease Kuzbanian to the apical domain to cleave Notch, likely through 3′ UTR-binding [42].

Regulation of gene expression in NCCs is another important process that relies on spatial regulation by RBPs. Gene function during neural tube formation and differentiation of NCCs is particularly important as defects in this process can affect overall tissue architecture and organism development [126,127]. The Wnt/β-catenin signaling pathway is a well-known regulator of cranial neural crest development, particularly epithelial-mesenchymal transition (EMT). NCCs undergo EMT during early development and errors in this process result in various abnormalities. Efforts to identify important components involved in NCC function revealed that EMT causes elevated levels of posttranscriptional regulators and ribosome biogenesis factors in chicken cranial neural crest cells [128]. Both are well known roles of RBPs, pointing to their importance in neural crest cell development and function. An example of this is the RBP Vera (Vg1) that is involved in RNA localization and migration of cells needed to form the neural tube and migration of NCCs in *Xenopus* oocytes [129]. Embryos lacking Vg1 still generate the appropriate cells, but they fail to migrate to their final tissue [129]. Similarly, the Hu/ELAV family of RBPs regulates migration of NCCs and timely expression of genes involved in neural crest specification in avian embryos [130]. Growth and NCC expansion are further regulated by the RBP cellular nucleic-acid binding protein (CNBP) [131]. *Xenopus laevis* embryos lacking CNBP had decreased levels of *foxD3* and *c-myc*, two important regulators of NCC growth and specification. Zebrafish forebrain development also employs a similar mechanism involving CNBP. In zebrafish embryos, CNBP promotes survival and regulates proliferation of NCCs [132]. Overall, the ability to transport and localize mRNA at a subcellular level plays an important role in gene expression during brain development. 

Brain development further relies on spatial gene expression within various neurogenic niches. These regions are dedicated to maintaining neural stem cell populations, gliogenesis, or repair following injury [133,134]. Along with their specific functions, these niches also express a specific subset of proteins [135]. For example, in the murine CNS, the RBP Quaking (mammalian: Qki, *C. elegans*: GLD-1, *D. melanogaster*: HOW, ZF: zqk) [136] regulates cell differentiation through regulation of alternative splicing, translation, and mRNA stability [137]. Qki binds the 3′ UTR of astrocytic mRNA [136] in the murine brain or it binds targets in a sequence-specific manner (CUAAC) in mammalian cells [138]. Its various isoforms are expressed in different areas of the CNS and in different cell types [139]. Within these tissues and cell types, Qki can localize to the cytoplasm or to the nucleus, depending on the isoform. This restricted expression is beneficial for the generation of astrocytes and oligodendrocytes from NSCs [140]. Qki expression can also be restricted to cells that co-express other factors. For instance, Qki5 and -6 are specific to neural stem cells that co-express the neural stem cell transcription factor paired type homeobox 6 (Pax6) [141]. In this way, Qki expression is limited to neural stem cells during early neurogenesis. Qki has additional functions that make it an important regulator of cell differentiation. Those additional functions are discussed in the Regulation of Temporal Gene Expression section below. Spatial regulation by RBPs is a highly conserved mechanism used in various tissues, and the function of Qki and related RBPs highlight an important role that these proteins play specifically in spatially distinct niches controlling neural development. 

RBPs can also participate in spatial regulation of gene expression by regulating the location of scaffolding proteins and their cognate mRNA. *Drosophila* neuroblast development involves precise localization of cellular machinery during mitosis to produce progeny with the correct fate. This is accomplished through establishment of cell polarity and asymmetric cell division. Both processes have been well studied in *Drosophila* neuroblasts, although some important details of underlying mechanisms remain unknown [142,143]. One well known model in *Drosophila* is localization of the scaffolding protein Miranda (Mir) to the basal domain of the neuroblast. This event signals for recruitment of important components required for differentiation. For example, targeting of the RBP Staufen (Stau) to the basal domain is accomplished by the scaffolding protein Miranda (Mir) [77]. Basal targeting of Stau and its bound RNAs [144] facilitates their asymmetric segregation during cell division, and thus, differential expression of cell-fate determining genes [145]. Stau functions early in *Drosophila* development to promote localization of bicoid and oskar RNA in the oocyte [146,147]. Further, Stau is required for basal localization of Prospero (Pros) RNA in mitotic neuroblasts—an event that is associated with production of differentiated cells [148,149]. Loss of Stau leads to improper localization of its target mRNAs, and thus, improper cell fate [145]. Overall, generation of diverse cell types in the embryonic and larval CNS in the fly is highly dependent on spatial gene expression that can be aided through assembly of protein scaffolds directed by intrinsic polarity cues. 

Targeting of RBPs and RNA to different subcellular regions establishes the apical and basal domains necessary to guide subsequent divisions. This mechanism is also observed in development of neural precursor cells and neural stem cells in the mammalian cerebral cortex. There are two Staufen genes in humans, Staufen1 and Staufen2 (Stau1/2). Stau1 is more ubiquitously expressed, whereas Stau2 is mostly expressed in the brain. Stau1 is recruited by the zinc-finger protein Kruppel-like factor 4 (Klf4) to the 3′ UTR of mRNAs involved in cortical neurogenesis. Once bound, Stau1 promotes degradation of these targets to maintain neural precursor cell (NPC) identity [150]. Apart from this, Staufen1 also functions in a complex with Barentsz (Btz) to localize mRNA, a function that is conserved in mammalian hippocampal neurons and *Drosophila* oocytes [151]. Unlike dStaufen, Stau1 is expendable during NPC development and self-renewal [152]. This is likely due to redundant roles between Stau1 and Stau2. During asymmetric cell division, Stau2 is enriched in the intermediate progenitor cell (IPC) where it regulates targets involved in mitosis, transport, and centrosome assembly [153]. Thus, conserved RBPs appear to play similar roles in neurodevelopment.

Spatial expression of RBPs is an efficient way to regulate the gene expression profile of cells and specialized tissues. This method allows for efficient reprogramming to establish cell identity—often observed during the progenitor to differentiated cell transition. These processes are important for tissue establishment during early development. Beyond this, maintenance of cellular identity and tissue homeostasis relies on proper functioning of these RBPs. For some RBPs, this means expression throughout several developmental stages, whereas other RBPs are expressed only during specific developmental windows.

## 4. Regulation of Temporal Gene Expression

RBPs can influence target metabolism through regulation of RNA splicing, translation, turnover, transcription, and modification. These functions can impact temporal gene expression, which refers to the expression of genes or genetic profiles during specific developmental windows. This property is conserved in many organisms including flies, worms, and zebrafish and assists in specification of cell identity. Temporal gene expression patterns in *Drosophila* govern the timing of transcriptional reprogramming and this mechanism is also conserved in *Caenorhabditis elegans* (*C. elegans*) and *Danio rerio* (zebrafish) [154,155]. Temporal gene expression in *C. elegans* regulates memory processing, including storage and retrieval [156]. Transient expression of certain transcription factors in *C. elegans* regulates gene activation in preparation for neuronal specification [157]. This mechanism is further conserved in mammalian neural stem cells in which the differentiation program is temporally regulated by lncRNAs and other components [158,159]. Translocated in liposarcoma (TLS) is an RBP that inhibits transcription in response to ncRNA signals. Cyclin D1 (*CCD1*) is a gene involved in cell cycle progression and neural stem cell proliferation [160]. In human cell lines, ncRNAs bind the 5′ region of *CCD1* to promote TLS-mediated repression [161]. Similar mechanisms are also in play in the mammalian spinal cord where temporal regulation aids in establishment of neuronal lineages [162]. Not only is the mechanism of temporal gene expression conserved across different species, but certain families of RBPs are also conserved. Mechanisms of temporal gene expression can occur by temporal target regulation by the RBP or via temporal expression of the RBP itself, and notable examples of such regulation are spotlighted in the following paragraphs. 

The RBP family of ELAV/Hu embryonic lethal abnormal vision (ELAV) proteins is a widely studied example of temporal gene expression. ELAV/Hu proteins have roles in synaptic plasticity, mRNA stability, differentiation, and nuclear functions [163]. These highly conserved proteins (Humans: *HuB*, *HuC*, *HuD*, *HuR*, Flies: *ELAV*, *fne*, *RBP9*, *C.elegans*: *exc-7*, ZF: *ELAVL1a*) are necessary for development of neurons that will populate the CNS (Table 1). As such, ELAV is expressed during neuron formation [164]. They have important roles in pre-mRNA splicing of nervous-system-specific isoforms, establishment of brain vasculature during early development, and gene expression in cholinergic motor neurons [63,69,165]. Regulation by ELAV/Hu occurs primarily through modification of target localization and expression levels [166]. For instance, HuD binds targets associated with plasticity and interacts with the survival motor neuron (SMN) protein to transport mRNA within mouse axons [167]. This is accompanied by changes in chromatin architecture to establish a gene expression profile consistent with a differentiated cell [168], which are particularly important during neural stem cell to neuron transition. Many of the specific roles of the ELAV family of proteins have been described [169,170,171,172], so we will not discuss them further here.

Expression of certain RBPs with opposing functions is another common mode of temporal gene expression. An example of this is the expression pattern of IGF-II mRNA-binding (Imp/IGF2BP2) and Syncrip (Syp) in *Drosophila* where they regulate neuroblast growth and termination [5]. Imp/IGF2BP2 and Syp have opposing expression levels throughout development. Both RBPs regulate target localization, stability, and translation [55,173]. Specifically, developmentally timed expression of Imp and Syp in *Drosophila* NSCs regulates cell fate and promotes neuronal diversity [5,174,175]. Imp is highly expressed during early development and suppresses expression of Syp. Imp regulates the mushroom body (MB) lineage in *Drosophila* by regulating expression of Chinmo mRNA. Chinmo is responsible for limiting neuroblast self-renewal [176], and thus, is an important regulator of cell specification. Imp is also a key regulator of neuroblast size and growth rate through interaction with Myc mRNA [175]. The Imp–Myc mRNA interaction stabilizes the transcript to increase Myc protein levels and promote neuroblast growth [175]. Apart from being important in early development of the brain, Imp and Syp are also important in halting brain growth by preventing future neuroblast divisions through initiation of timely decommissioning and differentiation. In contrast to Imp, Syp exhibits minimal expression during early development. Levels of Syp gradually increase and reach significant expression levels by later larval stages. This shift in Syp expression serves to inhibit Imp expression during later developmental stages [5]. This event marks the end of neurogenesis, and thus, is an important step in forming tissues of the proper size (see RBPs in disease and dysfunction, below). Additionally, Syp regulates late developmental stages through interaction with miRNAs. Recently, it was shown that Syp also interacts with the miRNA pri-let-7a to regulate the larvae–adult transition in flies and worms and to suppress tumors in mammalian cells (let-7) [177,178]. Syp is also involved in neuroblast differentiation through regulation of Pros mRNA with a long 3′ UTR. Binding of Syp to the extended 3′ UTR of Pros mRNA stabilizes the transcript to promote increased Pros protein production [179]. This isoform of Pros is not expressed earlier in development [180] but is necessary for production of larval neurons [179]. Similar mechanisms of mRNA regulation are observed in mammalian neurons where Syncrip regulates plasticity, neuronal development, and differentiation through 3′ UTR-mediated repression or stabilization of target mRNAs [181]. Thus, gene regulatory networks involving multiple RBPs and their respective targets can participate in temporal gene expression patterns that facilitate proper developmental transitions in neural stem cells.

Finally, alternative splicing offers yet another form of RBP-mediated temporal gene expression. RBPs are important mediators of alternative splicing in neural tissue during early and late development [182,183] (Figure 1). Protein variants and transcripts found in different neural cell types are often generated through alternative splicing [184]. For example, RNA-binding Fox-1 homolog 1 (Rbfox1) is itself alternatively spliced, and transcript isoforms containing specific exons are expressed in specific regions of the brain [185]. During the neural progenitor cell-to-neuron transition, Rbfox1 in turn ensures correct inclusion of neuronal exons during alternative splicing in the cerebral cortex [186]. Another example is the previously mentioned RBP Qki. Its roles in alternative splicing are important for NPC differentiation and, like Rbfox1, Qki isoforms are present in specific tissues and developmental stages [187]. Remarkably, Qki in oligodendrocytes autoregulates its own splicing events [188]. Qki5 has been shown to regulate targets in early embryonic neural stem cells. Axon development and microtubule dynamics were shown to be two processes dependent on Qki5 function [189]. Isoform specificity further extends to differentiated cell types such as neurons and oligodendrocytes [188]. For example, Qki promotes a GABAergic neuron profile, and loss of Qki alters the gene expression to promote a glutamatergic neuron profile [189]. A comprehensive review of the role of RBPs in neuronal differentiation can be found in [190]. In oligodendrocytes, mRNAs associated with myelination are significantly downregulated [188]. Interestingly, expression and function of the RBPs themselves are also regulated, with many RBPs being essential in certain tissues and stages of development. For instance, the RBP Hu antigen R (HuR) is dispensable during embryonic development but essential during adult neurogenesis [191]. Further, HuR regulates a group of lncRNAs involved in differentiation. More directly, HuR expression is necessary to maintain stemness and prevent aberrant increases in cells expressing neuronal markers [192]. Together, these examples point to the importance of regulating gene expression profiles to obtain cells with the desired properties. Still, not all temporal regulation occurs at the transcriptional level. Many mechanisms are in effect following transcription and these events are described in the next sections. 

## 5. Posttranscriptional Regulation by RNA-Binding Proteins 

Posttranscriptional regulation is an emerging means of maintaining stem cell homeostasis. This mechanism aids in the response to cellular and environmental cues throughout development. During early development, stem cells must undergo transcriptional remodeling to become differentiated cell types. This includes expression of the important protein factors, such as differentiated cell markers, and repression of stemness genes including growth promoting factors. The expansive repertoire of RBPs can aid in regulating this process through control of protein translation, a process particularly important in stem cells [193,194,195]. Apart from directly influencing target metabolism (splicing, localization, degradation, etc.), RBPs can also regulate targets indirectly. One such mechanism is regulation of microRNA function and biogenesis. This mechanism is common in brain development and recent work has highlighted the important role of miRNAs in regulation of growth and proliferation [196,197]. Moreover, the complexity of cell types in the brain has been attributed to wide expression of miRNAs [198]. Until recently, few RBPs were thought to regulate miRNA structure and function, with only a handful of RBPs being able to bind miRNAs. Further studies have showed that RBPs facilitate the miRNA–mRNA interaction. They do so by exposing binding sites that would otherwise be inaccessible by the miRNA (Figure 1). Pumilio does this by binding the 3′ UTR of *CDKN1B* to allow binding by its regulatory miRNA [199]. Conversely, RBPs can also inhibit binding site access, including the mammalian RNA-binding motif protein 38 (RBM38) and insulin-like growth factor 2 mRNA-binding protein 2 (Imp2) [200,201]. Apart from influencing the interaction between miRNA and its targets, RBPs can regulate miRNA processing. 

RBPs can also regulate targets by interacting with miRNAs or with targets that resemble the miRNA structure. One example of this is the RBP DiGeorge syndrome critical region 8 (DGCR8). This RBP is part of an important miRNA biogenesis component known as the microprocessor complex. DGCR8 and accompanying factors function to produce primary miRNAs (pri-miRNA), including some involved in neural stem cell growth [202,203,204]. For instance, this complex is responsible for processing of the pri-*miRNA-bantam*. This miRNA is part of a feedback loop with Numb and Notch to impart neural stem cell growth in the *Drosophila* brain [205]. The Drosha-DGCR8 complex is a well-known mechanism of miRNA processing. Details of this mechanism have been reviewed elsewhere [206]; thus, we will highlight the role of RBPs in miRNA processing and some of the miRNAs involved. RBPs have been shown to regulate gene expression through processing of miRNAs that act in a tissue-specific manner [202]. In addition to their roles in miRNA processing, DGCR8 and Drosha function to regulate expression of key transcription factors involved in murine embryonic neurogenesis. One of these factors is the transcription factor Neurogenin 2 (*Neurog2*). The Drosha–DGCR8 complex promotes degradation of Neurog2 mRNA to prevent differentiation [207]. Interestingly, this inhibition is independent of the canonical Microprocessor functions. Rather, it is facilitated by the structure of Neurog2 mRNA that resembles pri-miRNA [207]. Further, DGCR8 also functions in neural progenitor stem cell maintenance and cortical development [208]. Several of the miRNA targets regulate differentiation potential and proliferation of neural stem cells [209]. For instance, the 3′ UTR of p53 undergoes regulation by the miRNA miR-302. Consistent with this, cells depleted of miR-302 exhibit increased p53 activation that prevents cell differentiation [210]. Similar defects are observed when the RBP components of the miRNA processing machinery are misexpressed. Overexpression of DGCR8 in the murine cortex promotes an expansion of the neural progenitor cell pool, whereas loss of DGCR8 leads to apoptosis and defective corticogenesis [208,211]. Similarly, the RBP LIN28A/B, which is highly expressed in undifferentiated cells, prevents miRNA-dependent differentiation, and influences growth of neuroblasts (neuroblast: in fruit flies, these are the neural stem cells that produce glia and neurons) and neural progenitor cells (neural progenitor cell: cells with limited differentiation potential that produce specific neuron and glial subtypes) in mice [212]. As described, miRNAs have important roles in cell growth and differentiation. They provide additional means to maintain tissue homeostasis through their various functions across different tissues. RBPs regulate these various functions through miRNA processing and binding of miRNA to their targets. Thus, RBPs ensure that miRNA functions are carried out when it is developmentally relevant. 

Target metabolism is also regulated by RBPs through interference with translational machinery. For example, the fragile X mental retardation protein (FMRP) regulates protein translation through ribosome stalling [213]. In this way, translation of certain mRNAs is reduced, and this is central to the cellular response to signals or to adapting to a changing environment. Similarly, cells rely on ribosome stalling during cellular differentiation, namely, neuronal differentiation, where transcripts are regulated posttranscriptionally through 3′ UTRs rather than through mRNA levels [214,215]. Similar mechanisms are used during early murine forebrain development where the protein synthesis machinery is downregulated at later stages of forebrain establishment [216]. Translation can also be inhibited before the ribosomes are loaded onto the mRNA by acting directly on the target. Binding of the RBP to the transcript prevents ribosome loading, and thus, protein synthesis. Conversely, RBPs may enhance protein synthesis by promoting recruitment of the necessary machinery or by creating a favorable RNA structure, such as with *LIN28*-mediated effects on neural stem cell maintenance [217]. 

The RBP cold-shock domain-containing E1 (CSDE1) regulates targets at the posttranscriptional level to temper their stability and translation. Interestingly, CSDE1 can function at different levels of translation and can tailor its target metabolism according to the tissue or cell type [218,219]. More specifically, CSDE1 promotes a stem cell profile in human embryonic stem cells (hESC) to prevent neural differentiation. This is performed through negative regulation of targets involved in radial glial cell formation (RGC). Consistent with this, loss of CSDE1 results in precocious neurogenesis, whereas overexpression leads to impaired neurogenesis [220]. Similarly, the RBP CUGBP ELAV1 family member 1 (CELF1) has an important role in murine neocortical development by regulating production of glutamatergic neurons. This is accomplished through translational repression of ELAV4 through a mechanism involving the 5′ UTR-binding. Unlike previously discussed RBPs, CELF1 is involved in temporal regulation of different isoforms of target RNAs, namely, *ELAVl4*. That is, isoforms are differentially regulated in early and late corticogenesis, and this is determined by the 5′ UTR of the target mRNA [221]. The RBP Pumilio2 (Pum2) also acts posttranscriptionally to regulate mammalian neuronal differentiation. Pum2 does this by initiating translation of *elF4E* to ultimately promote neuronal growth [222]. This mechanism adds an additional level of complexity to RBP-mediated regulation of gene expression. 

Posttranscriptional regulation is an efficient mechanism that can be used for rapid changes in gene expression. Transitions from stem cells to more differentiated cell types must be accurate and rapid to keep up with a developing organism. More importantly, posttranscriptional regulation is a mechanism often observed during early development. For example, the *Drosophila* embryo relies exclusively on maternally inherited mRNA and proteins for the first three hours following fertilization. During this time, posttranscriptional regulation of these maternal genes and proteins guide developmental stages until transcription is turned on later in development [223,224,225]. Maternal-zygotic transition in *Xenopus* also depends on posttranscriptional regulation to degrade maternal transcripts during cell-fate determination [226]. Together with temporal and spatial gene expression, posttranscriptional mechanisms provide multiple layers of regulation. The benefits and malfunctions of these complexities will be discussed in the following sections. 

## 6. RBPs and Phase Separation

Cells contain microenvironments that facilitate biological processes including transcription, degradation, and translation. To separate the various processes happening concurrently and ensure fidelity, cells contain canonical membrane-bound organelles along with more recently appreciated membraneless structures. Formation of membrane-bound organelles ensures that components can be restricted to discrete locales, whereas membraneless condensates allow for more dynamic movement of relevant molecules, as well as asymmetric subcellular localization in some cases. Still, these membraneless structures must also be regulated to maintain cellular homeostasis. Membraneless condensates are increasingly understood to be formed and regulated through liquid–liquid phase separation (LLPS) [227]. This process results in compartments that resemble an emulsion of oil droplets and water [228]. The transient and less restricted nature of the resulting membraneless structures, termed biomolecular condensates, makes them useful for rapid changes, such as cell specialization. Biomolecular condensates provide an efficient environment to carry out biochemical reactions and compartmentalize certain activities. These dynamic structures can be made up of different proteins and molecules, depending on the cell type, including a growing list of RBPs and bound RNA. Further, they can be formed in different parts of the cell, including the nucleus, and their formation can result from various conditions including stress and cell growth [229]. Structural specifications, such as intrinsic disorder, drive LLPS. More information about protein structure and the propensity to undergo LLPS can be found in excellent reviews [9,230,231,232,233]. 

Subcellular compartments such as P-granules (*C. elegans*), stress granules, and ribonucleoprotein (RNP) granules play important roles in posttranscriptional regulation. As such, these granules contain RBPs that are responsible for recruiting and regulating targets [234], as well as many of the other functions mentioned thus far. Further, structural properties such as intrinsically disordered regions (IDRs) (Figure 2) of many RBPs favor LLPS, and thus, creation of these subcellular structures (Figure 2) [235]. RBPs in biomolecular condensates have recently been shown to regulate stem cell homeostasis. The RBP fused in sarcoma (FUS) is involved in RNA localization, transcription, splicing, and DNA repair [236]. It has a role in maintenance of neural stem progenitor cells (NSPC), specifically through cell cycle regulation [237] and progression (S, G2/M) [238,239]. FUS acts in transcriptional condensates formed via LLPS to influence gene expression through recruitment of transcriptional machinery [240]. Interestingly, downstream functions of FUS are phase separation dependent. Namely, FUS that does not undergo LLPS interacts with RNA, whereas FUS that phase separates is involved in DNA damage repair and chromatin remodeling [241]. Moreover, it is responsible for regulating early differentiation. FUS contains an N-terminal prion-like domain (PLD) that facilitates reversible LLPS. This process is required for FUS functions including DNA repair [242], and this is important during stem cell divisions to ensure genome integrity is preserved. Stem cells are further regulated by RNP granules that promote translation of select targets. These highly conserved structures sequester mRNA and are formed through LLPS. Imp/ZBP1 is one RBP found in RNP granules in the *Drosophila* brain where it resides inside RNP granules along with translationally repressed mRNAs. Release of Imp and mRNAs from the granule results in translational activation [243]. Conversely, binding of RNA to RBPs can also promote formation of biomolecular condensates. Binding of RNA to hnRNPA1, an RBP enriched in the murine CNS [244], promotes phase separation and incorporation into stress granules [245]. Stress granules house stalled mRNA and RBPs and are frequently observed during stem cell differentiation. Like other biomolecular condensates, stress granules are not static. They disassemble once stressful conditions have been resolved and mRNA translation is reinstated [246]. Together with other resident proteins and nucleic acids, RBPs found in biomolecular condensates regulate cell fate spatially and temporally. Although much work remains to fully unravel the diverse roles that LLPS likely plays in neurodevelopment, it is clear this process represents yet another dynamic mode of RBP-dependent regulation to neural stem cell function.

## 7. RBPs Aid in Maintaining Tissue Homeostasis

The previous sections highlighted the importance of regulating cell growth and proliferation to ensure proper tissue establishment. However, RBPs also play an important regulatory role to maintain tissue function and prevent disease. For instance, an important feature of stem cells that must be controlled throughout development is their ability to self-renew and differentiate. This fundamental feature helps determine the number and size of cells that comprise a tissue, with aberrant regulation of these often leading to several developmental diseases. For example, the YAP/TAZ (*Drosophila*; Hippo signaling) pathway has been implicated in regulation of tissue size and cell number through functioning of its effector YAP (Yorkie in *Drosophila*). In the *Drosophila* brain, neuroblast proliferation, quiescence, and overall brain size is modulated by the Hippo pathway [255,256]. In mammalian systems, YAP/TAZ/Yorkie dysfunction has been linked to aberrant tissue growth in both stem cells and epithelial tissue [256,257,258,259]. Thus, the Hippo pathway is an important regulator of tissue homeostasis, although the mechanisms that regulate the pathway itself remain unclear. Recently, regulation of the Hippo pathway has been attributed to posttranscriptional regulation and RBPs. Recent work showed that the ribonucleoprotein Hrb27C modulates the phosphorylation state of the effector proteins YAP/TAZ/Yorkie to promote growth [260]. Consistent with known roles of the Hippo pathway in growth and cell differentiation, loss of Hrb27C also results in decreased proliferation and aberrant structure of differentiated cells [260]. Loss of Hrb27C influences Yorkie-target gene expression; however, a molecular mechanism remains undefined. Regulation of Hippo also occurs through posttranscriptional control of Yki RNA. The RBP Rox8 prevents anomalous Yki signaling by promoting degradation of Yki RNA via two mechanisms. Rox8 binds the 3′ UTR of Yki mRNA (see [261,262]) to promote Yki mRNA decay. Rox8 also interacts with the microRNA miR-8 to recruit RISC to degrade Yki mRNA. A similar mechanism is observed when *Drosophila* cells are transfected with mammalian Rox8, TIAR [105]. In mouse embryonic stem cells, loss of TIAR promotes self-renewal [263]. Collectively, these studies illustrate the important role of RBPs in the regulation of an essential and evolutionarily-conserved cell growth pathway that mediates homeostatic control of tissue and organ size.

## 8. RNA-Binding Proteins in Dysfunction and Disease

The widespread role of RBPs in development also makes their dysfunction a common source of disease. Notably, RBP dysfunction in the CNS often presents as neurodegeneration or morphological defects [264,265]. Mutations resulting in altered function and RBP localization have been shown to lead to disease (Figure 2). Many neurodegenerative diseases are caused by protein aggression or mutations in genes that regulate autophagy. Protein aggregates result from errors in protein folding, denaturation, stress conditions, or due to age [266,267]. They contain a mixture of components including elevated amounts of RNA and are observed in numerous motor neuron disorders [268,269]. The RBPs FUS, TDP-43, hnRNPA1, and MATR3 have all been shown to aggregate and cause disease [270]. Recent work on the RBP Musashi (Msi) highlighted its role in Tau protein aggregation, commonly seen in Alzheimer’s disease. Interaction between Msi and Tau results in formation of aggregates and a similar response is observed when Tau interacts with TIA1 [271]. TIA1 promotes Tau phase separation and generation of toxic, oligomerized Tau [272]. Decreased TIA1 levels in an aberrant Tau background results in increased neuroinflammation [273]. Similarly, disease can also arise from dysfunction in regulatory pathways. Returning to the example of the Hippo, this pathway is also regulated by RBPs in several diseases including brain, liver, and pancreatic cancer pathogenesis [274,275,276]. In glioblastoma, levels of the RBP cytotoxic granule-associated RNA-binding protein (TIA1) (*Drosophila*: Rox8) (Table 1) increase following YAP knockdown to prevent cell invasion in U87 glioblastoma cells [276].

As another important example, Staufen, the RBP essential for neural stem cell development both in the fruit fly and in mammals, has been associated with diseased states including amyotrophic lateral sclerosis (ALS) and certain types of dementia. In one study, spinal cord samples from patients with ALS showed significantly elevated levels of Stau1 along with increased mTOR signaling [277]. Stau1 has been shown to inhibit autophagy through activation of mTOR [278]. In ALS, mTOR activation leads to stimulation of certain astrocytes that results in motor neuron toxicity and death. Many homeostatic mechanisms rely on autophagy to clear the cell of debris and toxic compounds. Thus, a disruption in the autophagy process contributes to neurodegeneration. Other sources of disease include incomplete development of tissue or tissue of the wrong size. Microcephaly and neural tube defects are two conditions caused by improper regulation of cell growth and proliferation. Loss of the RBP LIN28A leads to microcephaly in mice and a double deletion of LIN28A/B leads to neural tube defects. The underlying cause for these phenotypes is reduced protein synthesis resulting from defective ribosome biogenesis and translation [217]. Protein synthesis is an important process necessary to maintain cellular functions. As such, deviations from translational homeostasis lead to cognitive defects. Fragile X mental retardation protein (FMRP) is an RBP that is expressed in the brain where it regulates translation. Not surprisingly, loss of FMRP in adult neural stem cells results in decreased production of neurons and defective hippocampal processes [279,280]. Diseases resulting from defective FMRP signaling or absence include autism, intellectual disabilities, and a role in addiction [281,282,283]. Much of the disease is caused by over proliferation of adult neural stem cells at the expense of neurons [284].

## 9. Conclusions

Stem cells have the potential to generate diverse progeny from a small population of cells. Diversity of the progeny is attributed to several mechanisms including hormone signaling, transcription factor activity, and changes in cellular metabolism [285,286,287]. It is important to note that some metabolic enzymes have RNA-binding activity, and some metabolic processes can result in posttranslational modifications that in turn influence enzyme interactions with RNA [288]. This is an interesting avenue that will require further investigation. Complexity among distinct cell progeny is further increased by the varying proteome of each cell type. Stem cell regulation at the transcriptional level has been widely studied; however, posttranscriptional and translational regulation are more contemporary fields that warrant further studying. Notable among the outstanding research topics are the functions of RBPs in both normal and disease processes in brain development. Some RBP functions are associated with healthy tissues, whereas others, such as LLPS, have primarily been associated with disease. As discussed in this review, LLPS of TDP-43 causes disease, whereas that of FUS is critical for DNA repair. Characteristics governing disease-associated LLPS have been elucidated for a limited number of RBPs; however, details remain unknown for several others. For instance, LLPS of TDP-43 results in disease; however, aggresome formation also leads to disease [289]. Additional pathways that lead to generation of aggresomes have not been determined for other proteins known to be involved in brain diseases, including hnRNPA1 and FUS [290]. Furthermore, molecular characteristics such as disordered regions, phosphorylation, and amino acid composition and their role in spatial and temporal regulation of aggresome formation and LLPS remain to be explored [291,292]. From a pharmacological standpoint, these details could further efforts to identify drug targets beyond the conventional single protein strategies [293,294]. 

More well-known, but equally important, contributors to disease are errors in RNA processing. Brain diseases can also arise from errors in RNA splicing, localization, and translation. These include errors in processing of target RNA or expression/processing of the RBP transcript itself, as observed with FMRP in fragile X syndrome (FXS) [295]. The initial need here is to identify potential regulators of RBP expression. These may be tissue specific or ubiquitous, depending on the RBP. That is, expression of RBPs that are tissue-specific must be induced by upstream activators and these may be different, depending on the tissue. Large-scale identification and analysis of RBP targets will also aid in understanding the role of RBPs in disease. Recent advances in technology have made this possible [296]; however, these studies would be well complemented by follow up studies in tissues of interest.

Stem cells and their progeny have different molecular components and express varying transcriptional and translational regulatory proteins, including RBPs [297,298,299]. These genetic profiles differ according to cell function and resident tissue. Thus, it is important that cells have the appropriate gene expression pattern to ensure correct development. The previous sections outlined the important roles of RBPs in establishing cell identity and maintaining tissue homeostasis. The diverse roles of RBPs allow them to be involved in many cellular processes including translation, miRNA processing, transport, splicing, and others mentioned here. Additionally, numerous RBPs function in a complex to exert different effects on targets (i.e., localization). Further, RBPs can alter when and where targets are expressed. Spatial and temporal regulation of targets can be particularly important during early and late developmental stages (see the Regulation of Temporal Gene Expression section). These various functions, coupled with the numerous RBPs identified to date, make them important regulators of neurodevelopment. This review discussed some of the important mechanisms in neurodevelopment that involve RBPs, which have only recently been identified as important regulators of cell fate. Many questions remain open for future exploration, including identification of additional RBPs and their key RNA targets across model organisms. Moreover, unraveling additional gene regulatory networks that rely on RBPs for proper regulation of gene expression is needed.

## Figures and Tables

**Figure 1 jdb-10-00023-f001:**
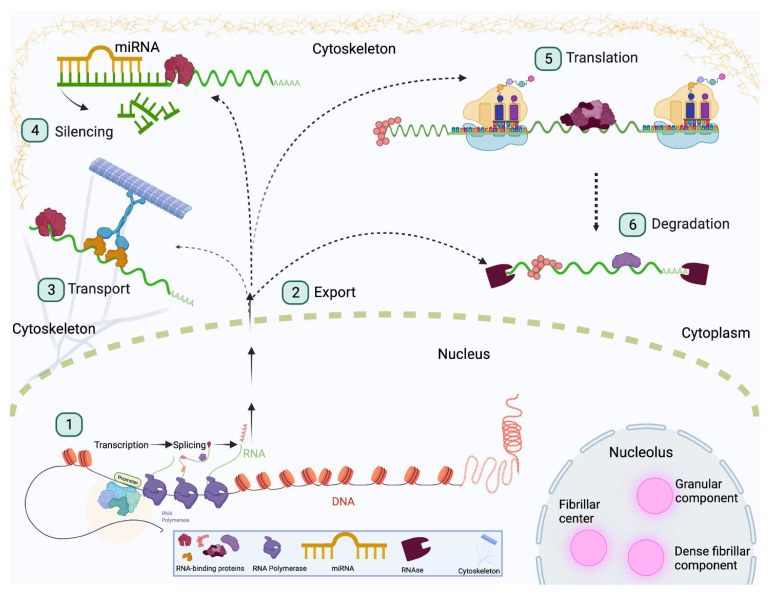
RNA-binding proteins regulate RNA fate. RBPs are involved in all stages of RNA processing including transcription, translation, splicing, transport, degradation, and silencing. These processes are depicted in the above diagram. RNA targets are bound by RBPs prior to leaving the nucleus and remain associated with RBPs until they are degraded. (1) RBPs such as Rbfox1 and Qki serve as alternative splicing factors to ensure cells have the correct genetic profile. (2) Shuttling of RNA between the nucleus and cytoplasm is also mediated by RBPs, as observed with CELF2. (3) RNA can also be transported to subcellular locations, and this is often conducted through a complex consisting of molecular motors and RBPs, as observed with ZBP1. (4) RBPs also interact with silencing factors such as miRNA to degrade certain targets. The RBP Pumilio facilitates miRNA binding by exposing binding sites that would otherwise be inaccessible. (5) RBPs also promote post-transcriptional processes. For example, Imp binds targets to stabilize mRNA and promote translation. (6) Some RBPs, such as Stau1, promote degradation of certain RNA targets to promote stem cell identity.

**Figure 2 jdb-10-00023-f002:**
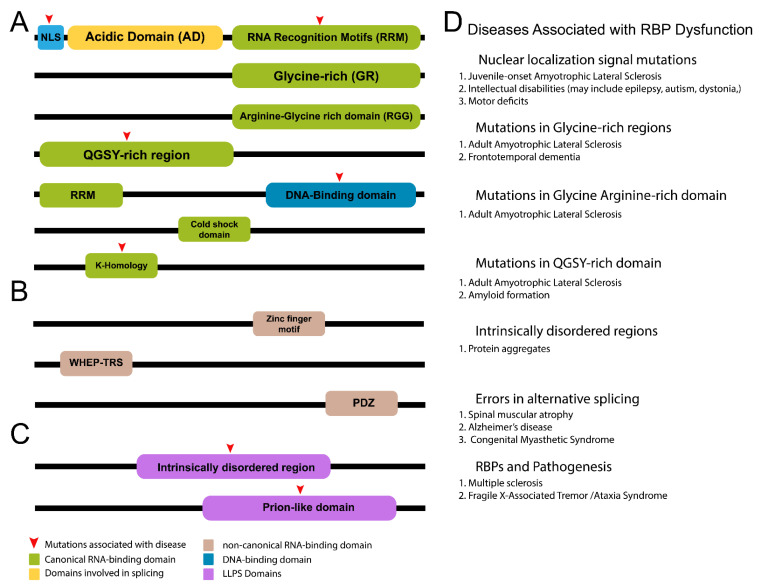
Structural Properties of RNA-binding proteins. Illustration depicts common RBP domains. (**A**) Most RBPs contain common RNA-binding domains, such as RRMs, GRs, and QGSY regions. Nuclear localization signals and acidic domains [247] (splicing factors) are also found in various RBPs. (**B**) Some RBPs contain noncanonical domains that can bind RNA/DNA [184], such as Zn-finger motifs, WHEP-TRS domains [248], and PDZ domains. (**C**) Other RBPs contain important regions involved in liquid–liquid phase separation that promote formation of biological condensates important for RBP function, such as stress granule formation and transcription. (**D**) List of RBP domains associated with disease and dysfunction. Common diseases and abnormalities are included [249,250,251,252,253,254].

**Table 1 jdb-10-00023-t001:** Conserved RNA-binding proteins. Table showing RBPs conserved in *Drosophila melanogaster* (Dmel), humans (Hs), *Caenorhabditis elegans* (Ce), and *Dario rerio* (ZF). Resident tissue, function, binding domains, and associated references for each RBP are also listed.

Protein Name	Tissue/Function	Binding Domain	Ref.
Dmel: IGF-II mRNA-binding protein (Imp) Hs: insulin-like growth factor 1/2/3 mRNA-binding protein (IGF2BP1/2/3) Ce: zipcode-binding protein 1 (ZBP1) ZF: insulin-like growth factor 2 mRNA-binding protein (IGF2BP3)	regulates stability, translation, transport of targets, axonal transport represses translation (IGF2BP2/3), 5′ UTR-binding (IGF2BP2), regulates cellular metabolism, interaction with miRNA, mRNA, lncRNA, germ cell maintenance translational repressor, 3′ UTR-binding embryonic and germline development, primordial germ cell migration, maternal mRNA stability	(4) KH domains, prion-like domain (PLD) KH domain, (2) n-terminal RRMs (4) c-terminal human heterogenous nuclear ribonucleoprotein (hnRNPs) (2) RNA-recognition motifs (RRM), (4) KH domains (2) RRMs, (4) KH domains	[41] [43,44,45,46,47] [48,49] [50,51]
Dmel: Syncrip (Syp) Hs: hnRNPQ/Syncrip Ce: HRP-2 ZF: synaptotagmin, cytoplasmic RNA-interacting protein (Syncrip)	mRNA regulation in neuromuscular junction, oocyte structure, neuronal fate in mushroom body neuronal RNA transport granules, translation, miRNA target regulation nucleic acid binding, embryogenesis, oogenesis, alternative splicing mRNA 5′ UTR-binding, synaptosomes, regulation of RNA translation	(3) RRMs, n-terminal unit for RNA recognition (NURR), NURR, arginine-glycine rich region (RGG), RRMs (3) RRMs (3) RRMs	[52,53] [54,55] [56,57,58,59] [54,60,61,62]
Dmel: embryonic lethal abnormal vision (ELAV) Hs: ELAV-like protein 4 (ELAVl4) Ce: EXC-7 ZF: ELAV-like RNA-binding protein 3 (ELAVl3)	alternative splicing, synapse formation, axon guidance, 3′ UTR extension binds AU-rich elements, 3′ UTR, translation, neuronal development, synaptic plasticity development of excretory canals, synaptic transmission, splicing, stability neurons, pan-neural marker, regulates alternative splicing, neuronal differentiation	(3) RRMs (3) RRMs (3) RRMs (3) RRMs	[63,64,65] [66,67] [68,69,70] [71,72,73]
Dmel: Staufen Hs: double-stranded RNA-binding protein Staufen homolog 1 (Stau1/2) Ce: Stau1 ZF: Stau1/2	enhanced translation, mRNA localization, cell fate, 3′ UTR-binding, ribonucleoprotein particles neuronal RNA transport (Stau2), mRNA decay (Stau1), memory formation, translation, 3′ UTR-binding (Stau1), double-stranded RNA-binding, germ cell development, miRNA interaction, elevated brain expression, primordial germ cells maintenance	(5) dsRNA binding domains (dsRBD), (1) proline-rich domain dsRBD (3′ UTR), microtubule binding, (1) proline-rich domain (1) proline-rich domain, (5) dsRBDs, (5) dsRBDs	[74,75,76,77,78,79] [80,81,82,83,84] [85,86] [87,88]
Dmel: Musashi1 (msi) Hs: Musashi1/2 (Msi1/2) Ce: Msi1 ZF: Musashib/Musashi2b (Msib/Msi2b)	adult external sensory organ development, asymmetric cell division (ACD), stem cell identity, translation, 3′ UTR-binding, sensory organ precursor cell ACD metabolism, stem cell self-renewal, cell cyle progression, binds 3′ UTR of mRNA, elevated in cancer cells (Msi1) Memory, learning, serotonergic signaling, 3′ UTR-binding, male mating behavior Expressed in neural tissue and progenitor cells, regulates cell proliferation and survival (Msi2b)	(2) RRMs (2) RRMs (2) RRMs (2) RRMs	[89,90,91,92,93] [90,94,95,96,97,98] [98,99,100,101] [102,103]
Dmel: Rox8 Hs: T-cell intracellular antigen-1-related protein (TIAR) Ce: TIAR1/2 ZF: TIA1 cytotoxic granule-associated RNA-binding protein-like 1	Expression of X-linked genes, alternative splicing, Yki mRNA decay, 3′ UTR-binding Translational silencing, primordial germ cell development Germ cell apoptosis, fertility, embryonic development, stress granule protein, inhibition of axon regeneration (TIAR2), RNA- and DNA-binding, stress granule component	(3) RRMs (3) RRMs (3) RRMs (3) RRMs	[104,105,106,107] [108,109,110,111,112,113] [114] [115]
Dmel: LIN28 Hs: LIN28A/B Ce: LIN28 ZF: LIN28A/B	symmetric stem cell division, cell growth, oogenesis, muscle formation, differentiation translational enhancer, inhibit miRNA expression, stem cell self-renewal cell proliferation, differentiation, pluripotency retina regeneration, early development	cold-shock domain, CCHS zinc-finger domains cold-shock domain, CCHC zinc-finger domains cold-shock domain cold-shock domain	[116,117,118] [119,120] [121,122,123] [124,125]

## Data Availability

Not applicable.

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
