# Peer review of "Emerging Roles of RNA-Binding Proteins in Neurodevelopment"

_jdb, 2022, doi:10.3390/jdb10020023_

Round 1
Reviewer 1 Report
This review is a good summary of work in the field and gives a comprehensive look at RBPs and their role in different aspects of neurodevelopment. The authors do a very clear review of the emerging role of RBP in neurodevelopment. They provide excellent context to the importance of these proteins and their functional roles in normal development and disease. The compiled table of conserved RBPs is particularly impressive and will serve as a great guide to the field.
In general the work is organized well. To improve the scope of the review a main suggestion is:
A) Adding some references for the role of RBPs in neural tube/CNS development and neural crest cells might add more value to the review. Towards that a few suggestions for references are added below
1) The Role of RNA-Binding Proteins in Vertebrate Neural Crest and Craniofacial Development;
doi: 10.3390/jdb9030034
2) RNA-binding proteins, neural development and the addictions;
DOI: 10.1111/gbb.12273
3) Neuron-specific cTag-CLIP reveals cell-specific diversity of functional RNA regulation in the brain
doi.org/10.1101/244905
B) It would also be helpful to have a few references or referral to several topics related to RBP which would add more value in terms of historical context or other important works in the field' I feel there are some missing references to other RBPs and some structural studies. Suggested references added below.
1) RNA-binding proteins and neural development: a matter of targets and complexes; DOI: 10.1097/00001756-200412030-00001
2) A large-scale binding and functional map of human RNA-binding proteins; https://doi.org/10.1038/s41586-020-2077-3
3) Pumilio2 Promotes Growth of Mature Neurons;
DOI: 10.3390/ijms22168998
4) An exon junction complex-independent function of Barentsz in neuromuscular synapse growth;
https://doi.org/10.15252/embr.202153231
5) The RNA-binding protein XSeb4R regulates maternal Sox3 at the posttranscriptional level during maternal-zygotic transition in Xenopus;
https://doi.org/10.1016/j.ydbio.2011.12.040
6) How RNA-Binding Proteins Interact with RNA: Molecules and Mechanisms; https://doi.org/10.1016/j.molcel.2020.03.011
7) Induced ncRNAs allosterically modify RNA-binding proteins in cis to inhibit transcription;
https://doi.org/10.1038/nature06992
Author Response
We sincerely appreciate the helpful comments from both reviewers. We have incorporated these suggestions in our revised manuscript as described below. We believe the changes strengthen our manuscript and hope the reviewers agree that we have satisfied all of their concerns.
Changes to address comments from Reviewer:
- We have included a discussion of neural crest cell development and function along with suggested references as suggested by the reviewer and feel this improves the scope and reach of our review. These changes can be found on lines 83-95 and 144-164.
- We have included a discussion of additional RBPs and some historical aspects that can also be found on lines 238-240 ,268-271 and 343-344.
- We have included a citation for the role of RBPs in addiction on line 598.
Again, we are grateful for these excellent comments and for both reviewers’ rapid and thorough analysis of our manuscript.
Reviewer 2 Report
Review of a manuscript “Emerging Roles of RNA-Binding Proteins in Neurodevelopment” by Parra and Johnston submitted to JGB.
The manuscript reviews recent advances in developmental neurobiology that contribute to our knowledge of the complex processes involved in brain development. The authors pay a special attention on the roles played by RNA-binding proteins in neural stem cell function and maintenance. The topic is very important and the review is timely. It will be interesting to the readership of JGB. The manuscript is very well written and will be beneficial for basic researcherf and neurologists. The following corrections and additions should be done.
Abstract
Line 17. “Altogether, these allow RBPs to influence gene expression to regulate various cellular processes.” This is an awkward sentence. First, it is unclear what means here “these”. Second, it is unclear why the authors consider the effect of RBPs on gene expression only.
Figure 1.
miRNA on this Figure is disproportionally large, its size should be decreased.
Conclusion. Line 514 “Protein aggregates result from errors in protein folding, denaturation, stress conditions, or due to age [247]. The authors should add here the citation of an article “Conformational diseases: looking into the eyes. Brain Res Bull. 2010 Jan 15;81(1):12-24. doi: 10.1016/j.brainresbull.2009.09.015. PMID: 19808079.”
Line 569. “The diverse roles of RBPs allow them to be involved in many cellular processes including translation and even miRNA processing.” It is not clear why the authors have selected these two processes. It will be beneficial if the authors present more cellular processes in which RBP play important role.
Conclusion. It will be valuable if the authors outline future directions which will be important for better understand the role of RBP in neural stem cell function and their role in brain diseases.
Conclusion. Line 557. “Diversity of the progeny is attributed to several mechanisms including hormone signaling, transcription factor activity, and changes in cellular metabolism” The authors should ex[lain how changes in cellular metabolism affect diversity of the progeny.
Author Response
We sincerely appreciate the helpful comments from both reviewers. We have incorporated these suggestions in our revised manuscript as described below. We believe the changes strengthen our manuscript and hope the reviewers agree that we have satisfied all of their concerns.
Changes to address comments from Reviewer:
- We have made suggested changes in the abstract to clarify the sentence regarding RBP function. These changes are on lines 17-18.
- We have also included suggested changes pertaining to protein aggregates on line 563.
- Lines 644-645 now include more cellular processes as suggested by the reviewer.
- The conclusion paragraph now has a brief discussion on RBPs and their role in brain diseases, and we thank the reviewer for suggesting this improvement.
- We appreciate the opinion on the size of Figure 1 components, but we feel they are ideal to accurately depict the importance of the miRNA-RNA without distracting from the functions depicted in other panels interaction. As such, we have not made any changes to this figure.
Again, we are grateful for these excellent comments and for both reviewers’ rapid and thorough analysis of our manuscript.